# TADIS: STEERING MODELS FOR DEEP-THINKING ABOUT DEMONSTRATION EXAMPLES

## ABSTRACT

Instruction tuning has been demonstrated that could significantly improve the zero-shot generalization capability to unseen tasks by an apparent margin. By incorporating additional context (e.g., task definition, examples) during the fine-tuning process, Large Language Models (LLMs) achieved much higher performance than before. However, recent work reported that delusive task examples can achieve almost the same performance as correct task examples, indicating the input-label correspondence is less important than previously thought. Intrigued by this counter-intuitive observation, we suspect models have the same illusion of competence as humans. Therefore, we propose a novel method called TADIS that steers LLMs for Deep-Thinking about demonstration examples instead of merely seeing. To alleviate the illusion of competence of models, we first ask the model to verify the correctness of shown examples. Then, using the verification results as conditions to elicit models for a better answer. Our experimental results show that TADIS consistently outperforms competitive baselines on in-domain and out-domain tasks (improving 2.79 and 4.03 average ROUGLE-L on out-domain and in-domain datasets, respectively). Despite the presence of generated examples (not all of the thinking labels are accurate), TADIS can notably enhance performance in zero-shot and few-shot settings. This also suggests that our approach can be adopted on a large scale to improve the instruction following capabilities of models without any manual labor. Moreover, we construct three types of thinking labels with different model sizes and find that small models learn from the format of TADIS but larger models can be steered for Deep-Thinking.

## 1 INTRODUCTION

Recently, instruction tuning (IT) has attracted much attention in the NLP community, which has shown effectiveness in improving the zero-shot generalization capability and gradually boosting performance on unseen tasks as the number of training tasks increases (Chung et al., 2022; Ouyang et al., 2022; Sanh et al., 2022; Taori et al., 2023). Researchers have explored efficient approaches to generate instructions (Wang et al., 2022; 2023; Xu et al., 2023), resulting in dozens of powerful large language models (LLMs) (Ouyang et al., 2022; Sanh et al., 2022; Chung et al., 2022; Taori et al., 2023). However, LLMs still struggle to follow the instruction precisely in some scenarios (Li et al., 2023; AlShikh et al., 2023), which hinders the applications and alignment of LLMs.

When applying instruction tuning with examples on LLMs, previous work provides either all positive examples (Ouyang et al., 2022) or specified positive/negative examples (Wang et al., 2022). SuperNI (Wang et al., 2022) adopted *task definition + positive examples + negative examples* format to facilitate the instruction understanding, which achieves significant improvement compared with the zero-shot setting. However, Kung & Peng (2023) found that delusive task examples (i.e., have correct input-output formats but incorrect input-output mappings) could also achieve almost the same performance as correct task examples. This conclusion reminds us of a well-known cognitive bias in psychology, called **illusion of competence (Kruger & Dunning, 1999): an illusion of competence occurs when someone believes they have learned something, but they have not.** For example, a student understands the solution for one problem when the teacher explains. However, he may fail to solve the same problem after a few days. We suspect models have the same illusion of competence as humans that they tend to merely learn the format instead of thinking when the correctness of examples is explicitly presented. Motivated by this, we introduce **TADIS** (**T**hink **A**bout

**D**emonstrations **I**nstead of merely **S**eeing), a novel method to steer models for thinking with classifying rather than just seeing. In this way, models are encouraged to learn the difference between positive and negative examples but not only the surface format. The schematic of our proposed method is illustrated in Figure 1. Specifically, we first replace the correctness of examples with only ordinal numbers (i.e., example 1 and example 2) and then the model is expected to predict the correctness of provided examples. Ultimately the model uses previous thinking and actions as conditions to elicit a better answer.

Extensive experiments show the effectiveness of TADIS in zero-shot and few-shot settings, outperforming baselines with 2.79 and 4.02 average ROUGLE-L on out-domain and in-domain datasets respectively. When positive or negative examples are unavailable for datasets other than SuperNI, we evaluate the performance with generated examples by Self-instruct (Wang et al., 2023). In this case, we observe that TADIS still consistently surpasses traditional methods in zero-shot and few-shot settings, which indicates the potential of TADIS as a better instruction tuning strategy even on a large scale without any manual labor. Moreover, we also analyze the performance of TADIS with different model sizes (770M and 3B). The results indicate that small models learn from the format of provided examples, while large models can be steered for careful thinking.

Our contributions are summarized as follows

- We propose a novel method called TADIS to steer models for thinking about demonstration examples instead of merely seeing. TADIS outperforms competitive baseline consistently across in-domain and out-domain datasets.
- TADIS can be combined with other methods of generating instruction-tuning data to further improve the model's instruction-following capabilities without any manual labor. The effectiveness of TADIS with generated examples shows that our method can be adopted on a large scale scenario. [1]

## 2 RELATED WORK

### 2.1 INSTRUCTION-TUNING

Instruction-tuning, a method fine-tuning large language models on multi-tasks with meta-instruction ahead (Wei et al., 2022). The effectiveness of instruction tuning has been proved in many prior works, which significantly improves the zero-shot generalization capability by a huge margin and gradually improves on unseen tasks as the number of training tasks increases. (Chung et al., 2022; Ouyang et al., 2022; Sanh et al., 2022; Taori et al., 2023) Most existing work focuses on data generation, data mixing and data format of instruction tuning. Self-instruct (Wang et al., 2023), a semi-automated process for instruction-tuning the pretrained LM using instructional signals from the model itself. Evol-Instruct (Xu et al., 2023) creates large amounts of instruction data with varying levels of complexity, which starts with an initial set of instructions and then asks LLMs to rewrite them step by step into more complex instructions. Mukherjee et al. (2023) develop Orca model, which captures not only surface-level responses from LLMs but also complex reasoning signals. Specifically, they guided LLMs to respond to reasoning intensive FLAN instructions with a series of predefined system prompts (e.g., "think step-by-step and justify your response"), spurring LLMs (e.g., GPT4). Wang et al. (2022) introduce SuperNI, a benchmark that covers 76 distinct task types of 1616 diverse NLP tasks, including but not limited to classification, extraction, infilling, sequence tagging, text rewriting, and text composition. Each task contains *a task definition* (a high-level description of the input and output of the task) and *demonstration examples* (some input-output examples for the task. e.g., positive and negative examples.). Cao et al. (2023) propose Instruct Mining, a linear rule for evaluating instruction-following data quality. Wu et al. (2023) propose a step-by-step instructions training method, which can help language models decompose the tasks and provide specific procedures for completing the target tasks. Chen et al. (2023b) finetuned the LLaMA on only 9k high-quality data filtered from the 52k Alpaca data by ChatGPT judging, which significantly outperforms the original Alpaca. Yin et al. (2023) conducted a detailed ablation analysis to understand which parts of a task definition are most important and found that output content is more important than input content, especially label information. However, these works don't

---

[1]Our code and models will be made public.

pay attention to *demonstration examples* in the instruction and even ignore the example information directly. We regard this tactic as an information loss. Probably the most relevant work is Kung & Peng (2023), they found that models trained with delusive task examples can achieve almost the same performance as correct task examples, which indicates the model learns the input-output format instead of input-output mapping. This conclusion is counter-intuitive for humans. We suppose that is because the model tends to merely see instead of thinking when the correctness of examples is explicitly presented. Therefore, we propose TADIS method to steer the model for Deep-Thinking about provided examples. Our experiments have proved our previous surmise (see Section 4.2).

## 2.2 IN-CONTEXT LEARNING

In-context learning (ICL), a prompt method without training, encourages the language models to learn from a few input-output examples (Liu et al., 2022; Rubin et al., 2022; Min et al., 2022a). Inspired by the promising performance of ICL, many studies have explored how to further improve standard ICL. Min et al. (2022b) and Chen et al. (2022) introduce meta-learning to better adapt LMs to ICL. Zhao et al. (2021) estimate models' bias towards each answer and then develop contextual calibration to adjust the model's output probabilities which not only improves accuracy but also reduces variance. Kim et al. (2022) propose SG-ICL that generates demonstration examples for in-context learning from LM itself instead of manual labeling. Active Prompting (Diao et al., 2023) selects the most uncertain questions as demonstration examples to further improve performance. Min et al. (2022c) find that replacing gold labels with random labels only marginally hurts performance, which indicates models learn the format rather than input label pairs. Yoo et al. (2022) revisit previous findings of Min et al. (2022c) and introduce novel metrics to prove that the input-label correspondence plays a more significant role in contextual demonstration than previously considered. However, most of these methods focus on the inference stage and explicitly show the correctness of the demonstration examples, which may result in mere seeing rather than learning. Our work focuses on the training stage. Inspired by Yoo et al. (2022), we conjecture that facilitating the model to learn the input-output mapping in the training stage is also important.

## 3 METHOD

The proposed TADIS method aims to encourage the model to generate answers based on the thoughts about few-shot examples instead of the mere observation. Specifically, the model is spurred to think about the correctness of the provided examples (i.e., positive or negative) first and then elicited for answering with the thinking as conditions. The model is expected to better digest the provided examples through a simpler classification task, which in turn helps for a better answer.

For each given task, we denote the task definition as $S_T$. Following SuperNI, each task has a training dataset $D = \{(X, Y)\}$ and an example pool consisting of positive and negative examples $S_E = \{(X^e, Y^e, L^e)\}$. For each input-output instance pair $(X, Y)$ in $D$, we randomly select $k$ examples from the example pool as $S_E^k = \{(X_i^e, Y_i^e, L_i^e), i \in [1, k]\}$. The $i$-th example consists of the input $X_i^e$, the output $Y_i^e$ and its correctness label $L_i^e$ (i.e., positive or negative). Unlike SuperNI, we replace the correctness label of examples with ordinal numbers in the prompt and split $S_E^k$ into two parts $M_E^k$ and $R_E^k$. Here, we call $M_E^k$ as *masked examples* and $M_E^k = \{(X_i^e, Y_i^e), i \in [1, k]\}$. $R_e^k$ is the corresponding correctness thinking results which are created from the correctness labels $\{L_i^e, i \in [1, k]\}$ using a sentence template (as shown in Figure 1).

TADIS consists of two stages: **Thinking** and **Answering**.

**Thinking** During the thinking stage, the model will predict the correctness of each provided example based on the task description. We use the next token prediction as the training objective to facilitate the learning from provided examples. Formally, the loss can be represented as:

$$\mathcal{L}_{\textbf{think}} = - \sum_{(X,Y) \in D} \log(P(R_E^k, A_E | S_T, X, M_E^k; \theta)). \tag{1}$$

Besides the correctness prediction, the model is also expected to generate the corresponding actions $A_E$ after thinking (e.g., "I should learn from correct examples and avoid mistakes in the wrong

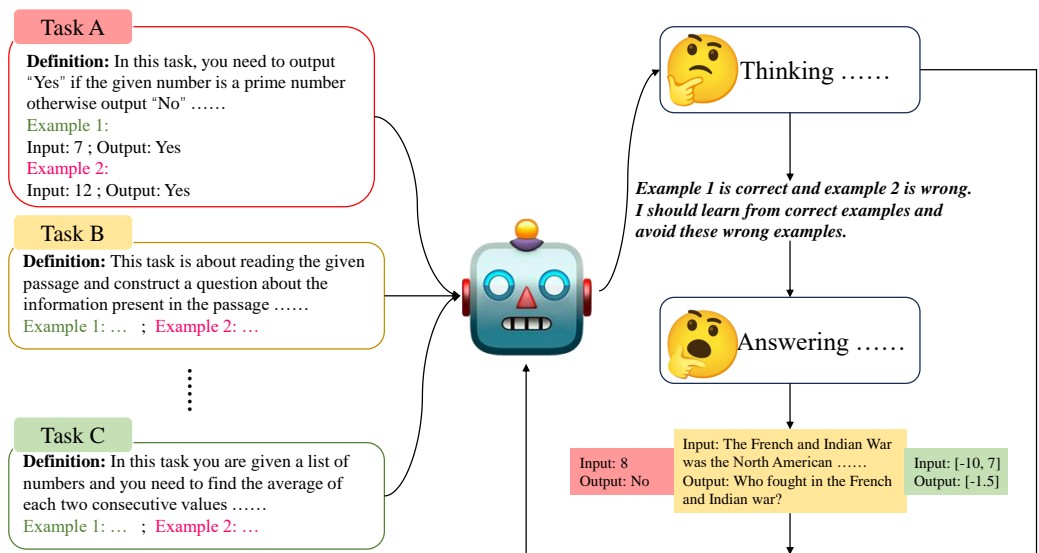

Figure 1: The overview of TADIS. TADIS consists of two stages: Thinking and Answering. (1) **Thinking:** Encouraging the model to judge the correctness of the examples to steer it for Deep-Thinking. (2) **Answering:** Using the thinking as conditions to elicit the model for a better answer. Two stages are executed sequentially rather than separately.

examples."). The action generation encourages the model to take the corresponding actions for better answers. We conduct further exploration and analysis in Section 4.3.

**Answering**     Based on the thinking result $R_E^k$ and the corresponding action $A_E$, the model continues to generate the answer $Y$ for instance input $X$. Similar with the thinking stage, the training loss of the next token prediction task is calculated as

$$\mathcal{L}_{\mathbf{answer}} = - \sum_{(X,Y) \in D} \log(P(Y|S_T, X, M_E^k, R_E^k, A_E; \theta)). \tag{2}$$

The overall training loss is the sum of these two losses $\mathcal{L} = \mathcal{L}_{\mathbf{think}} + \mathcal{L}_{\mathbf{answer}}$.

During inference, the model generates the answer after its thinking on the provided examples:

$$P(Y|S_T, X, M_E^k) = \sum_{R_E^k, A_E} \underbrace{P(R_E^k, A_E|S_T, X, M_E^k)}_{\text{Thinking}} \times \underbrace{P(Y|S_T, X, M_E^k, R_E^k, A_E)}_{\text{Answering}}, \tag{3}$$

It is prohibitively expensive to compute the sums due to the exponential search space of the thinking. In practice, we use the best thinking predicted by the model for approximation.

## 4 EXPERIMENTS AND ANALYSES

### 4.1 EXPERIMENT SETTING

**Dataset**    We conduct experiments on the SuperNI-V2 (Wang et al., 2022), the largest open-source instruction dataset, including over 800+ English tasks with diverse task types. For each task, it contains *Task Definition, Positive Examples, Negative Examples* and *Explanations*. We use the same split of the dataset as that of SuperNI: training set has 756 diverse tasks and test set contains 119 unseen out-domain tasks as held-out. Meanwhile, we also

Table 1: Statistics of our training, held-in, and held-out datasets.

| Statistics | Training datasets | Held-In | Held-Out |
|---|---|---|---|
| # Total Tasks | 756 | 756 | 119 |
| # Total instances | 45360 | 11340 | 11900 |
| # Avg examples | 1.83 | 1.79 | 1.75 |

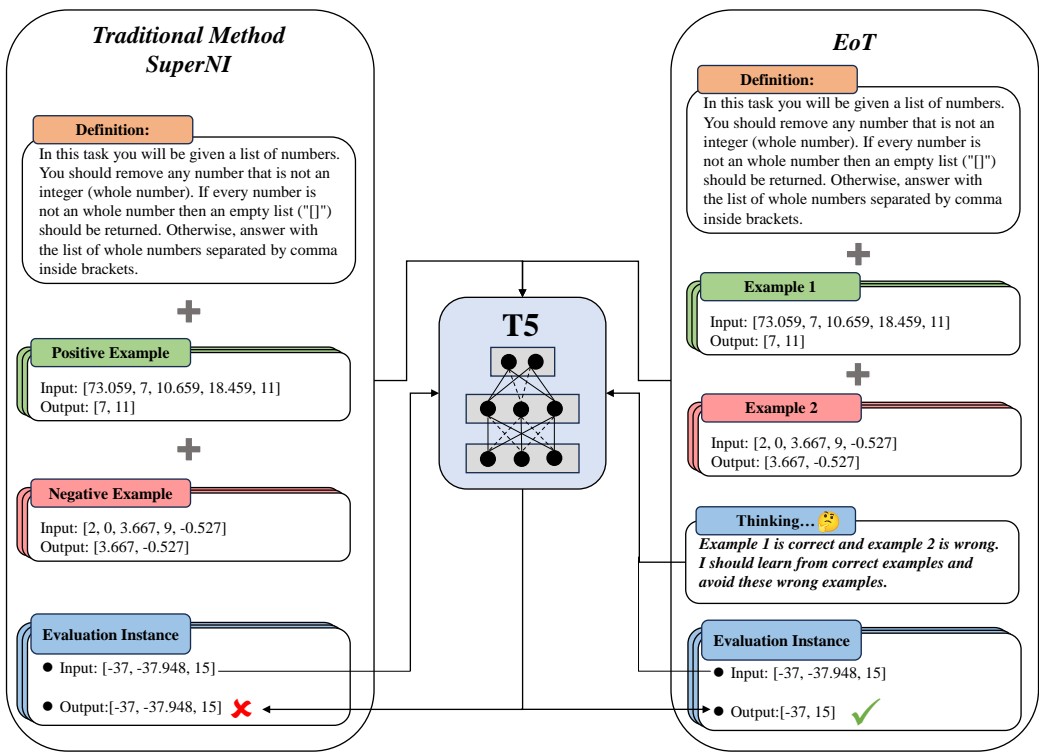

Figure 2: An example of comparing TADIS and traditional methods from `T5-xl-LM-Adapt`. Thinking before answering can elicit the model to the correct answer. However, the model cannot follow the instruction without Deep-Thinking.

construct a held-in dataset that contains the same tasks as the training set but with different evaluation instances to prevent leaking. As the performance saturates when the number of instances per task increases (Wang et al., 2022), we randomly select 60 instances for each task in the training set and 100 instances for each out-domain task in the held-out test set. For held-in dataset, we randomly select 15 instances for each in-domain task to maintain a similar number to the held-out dataset. The statistics of our training, held-in and held-out datasets are presented in Table 1.

**Baseline and Metrics**  We utilize `T5-LM-Adapt` as our backbone model following Kung & Peng (2023). Specifically, we explore TADIS on two different sizes of `T5-LM-Adapt`, i.e., `T5-Large-lm-adapt` (770M parameters) and `T5-XL-lm-adapt` (3B parameters). During the inference phase, we follow Wang et al. (2022) to sample from model outputs via greedy decoding (i.e., set temperature to 0) to obtain the most confident answers. We consider both zero-shot and few-shot settings in the training and inference phases. For the zero-shot setting, we use *Definition + evaluation instance* as a sample. For the few-shot setting, we use *Definition + Positive examples + Negative examples + evaluation instance* as a sample. Both settings remain the same as that of SuperNI and we also employ this traditional training format as our baseline. TADIS only has few-shot setting in the training phase. Due to the diversity of tasks and the open-ended generation nature of formulation, we adopt ROUGE-L metric for reporting aggregated performance results, which are observed to correlate well with accuracy for classification tasks and human evaluation (Wang et al., 2022).

## 4.2 EXPERIMENTAL RESULTS

To prove the effectiveness of TADIS, we compare the performance of TADIS with SuperNI (zero-shot) and SuperNI (few-shot) methods on the held-in and held-out datasets, respectively. SuperNI (zero-shot or few-shot) means the model is trained with zero-shot or few-shot settings. The training set was kept constant for all three methods except the format. However, we restricted the maxi-

Table 2: The performance (ROUGE-L) under zero-shot and few-shot settings for three methods on hold-in and hold-out datasets. **Avg RougeL:** We calculate the average of ROUGE-L under zero-shot and few-shot settings. **Bold** denotes the best result.

| Model | #Params | Testing Setting → Training Setting ↓ | Held-Out | | | Held-In | | |
|---|---|---|---|---|---|---|---|---|
| | | | Zero-Shot | Few-Shot | Avg RougeL | Zero-Shot | Few-Shot | Avg RougeL |
| T5-Large-LM-Adapt | 770M | SuperNI(Zero-Shot) | **38.016** | 40.59 | 39.30 | **46.22** | 42.59 | 44.40 |
| | | SuperNI(Few-Shot) | 33.30 | 45.08 | 39.19 | 43.59 | 52.96 | 48.27 |
| | | TADIS | 33.59 | **46.67** | **40.13** | 44.67 | **53.31** | **48.99** |
| T5-XL-LM-Adapt | 3B | SuperNI(Zero-Shot) | 42.89 | 45.73 | 44.31 | **49.95** | 47.59 | 48.77 |
| | | SuperNI(Few-Shot) | 38.54 | 51.08 | 44.81 | 41.49 | 52.96 | 47.23 |
| | | TADIS | **43.09** | **52.11** | **47.60** | 47.29 | **55.21** | **51.25** |

mum number of provided examples to two for both SuperNI (few-shot) and TADIS. The results are represented in Table 2. TADIS consistently outperforms SuperNI (zero-shot) and SuperNI (few-shot) methods on the held-in and held-out datasets. Moreover, the larger the model, the greater the benefit is. Specifically, for a larger `T5-XL-LM-Adapt` (3B) model, the performance combined with TADIS improves by 2.79 and 4.02 average ROUGEL-L in zero-shot and few-shot settings on held-out and held-in datasets, respectively. The smaller model(`T5-Large-LM-Adapt`) only demonstrated a marginal increase of 0.94 and 0.72 average ROUGEL-L improvement. We suppose that is because larger models have stronger learning capabilities and can benefit more from TADIS methods. It is also worth noting that TADIS can improve more on held-in than held-out datasets (4.02 vs 2.79), which shows TADIS can also significantly benefit seen tasks. The model performs better with TADIS on in-domain tasks since it has learned from the held-in dataset tasks in the training phase. In the zero-shot inference setting, SuperNI (zero-shot) method achieves the highest performance. Nevertheless, the performance drops sharply in the few-shot setting. This occurs because training and inference are harmoniously aligned, improving performance with consistent formats. In conclusion, TADIS benefits the performance in both zero-shot and few-shot settings and achieves the highest average ROUGE-L compared with baselines.

## 4.3 ANALYSES

**Does TADIS steer models for Deep-Thinking or just format learning?** To explore whether TADIS steers the model for Deep-Thinking, we set up three different thinking labels in training: **Ground Truth**, **Random** and **Flip**.

- **Ground-Truth:** A standard TADIS that we train the model with true thinking labels.
- **Random:** We randomly select thinking labels to replace true thinking labels.
- **Flip:** We flip all thinking labels. For example, thinking: *Example 1 is correct and example 2 is wrong*, we flip it to *Example 1 is wrong and example 2 is correct*. And the corresponding actions also need to be flipped.

Table 3 shows the results. In addition to the standard TADIS inference setup that generates thinking from the model (**Generated**), we also add these three types of thinking for comparison. We can observe that the small model achieves almost the same performance on Ground-Truth, Random and Flip training settings, which suggests TADIS steers model for format learning instead of thinking. However, for the larger model, the performance consistency decreases when we replace true thinking labels with random thinking labels. The phenomenon introduces an interesting conclusion: **small model learns the format of TADIS, but TADIS can steer larger model for real Deep-Thinking and both model sizes benefit from TADIS method.** We suppose that the Deep-Thinking may be an emergent ability for a large model. Meanwhile, for the flip training format, the performance doesn't drastically drop as the random training format. We suspect that we actually show the model with true thinking labels when we flip the labels, as only two thinking labels (i.e., positive and negative) are used in this case. Furthermore, we notice that the performance doesn't increase or decrease even if giving the model ground-truth, flip or random thinking labels for both small and large models

Table 3: The Performance (ROUGE-L) of TADIS with different thinking labels in the training and inference stages. We compare four different thinking labels for few-shot examples during inference, namely, **Generated** thinking results from the model, **Ground-Truth** thinking results, **Flip** thinking results and **Random** thinking results.

| Model | #Params | Testing Setting → Training Setting ↓ | Zero-Shot | Few-Shot | | | |
|---|---|---|---|---|---|---|---|
| | | | | Generated | Ground-Truth | Flip | Random |
| T5-Large-LM-Adapt | 770M | TADIS (Ground-Truth) | 33.58 | 46.66 | 46.67 | 46.68 | 46.72 |
| | | TADIS (Random) | 34.23 | 46.17 | 46.10 | 46.02 | 46.11 |
| | | TADIS (Flip) | 34.47 | 46.10 | 46.07 | 46.09 | 46.05 |
| T5-XL-LM-Adapt | 3B | TADIS (Ground-Truth) | 43.09 | 52.11 | 52.17 | 52.14 | 52.07 |
| | | TADIS (Random) | 33.52 | 45.76 | 46.14 | 46.02 | 46.11 |
| | | TADIS (Flip) | 38.95 | 51.25 | 51.31 | 51.29 | 51.30 |

in the inference stage, which indicates that TADIS steers the model for Deep-Thinking and format learning in the training stage rather than the inference stage.

**What's the relationship between judgment accuracy and performance in the training phase?** To further validate that the example verification process could steer model for Deep-Thinking, we calculate the correctness of judgment and ROUGE-L during the training phase with `T5-XL-LM-Adapt` for five epochs. The results are shown in Figure 3. We observe that judgment accuracy is extremely related to ROUGE-L through the slope. The Pearson correlation coefficient achieves a high value of 0.98, suggesting judgment accuracy and ROUGE-L are significantly correlated. Overall, the detailed statistics further demonstrate that forcing models to judge the correctness of provided examples could steer them toward Deep-Thinking.

**Can TADIS still benefit the performance with generated examples?** Considering the scarcity of negative examples in real application scenarios, we explored the performance of TADIS with generated examples. Specifically, we use Self-instruct (Wang et al., 2023), a framework for improving the instruction-following capabilities of pretrained language models by bootstrapping off their own generations. We choose the ChatGPT (`gpt-3.5-turbo-0613`) as our backbone LLM to generate new positive and negative examples for each task. We randomly selected eight pairs of positive and negative examples from different tasks to form our few-shot pool. For each generation, we construct the prompt with task definition and few-shot to generate new pairs of positive and negative examples. To generate diverse examples, we randomly select four (Chen et al., 2023a) pairs of positive and negative examples from the pool and shuffle their order. Meanwhile, we also set the temperature to 0.7 to further improve the diversity. Due to the API cost, we only constructed 5040

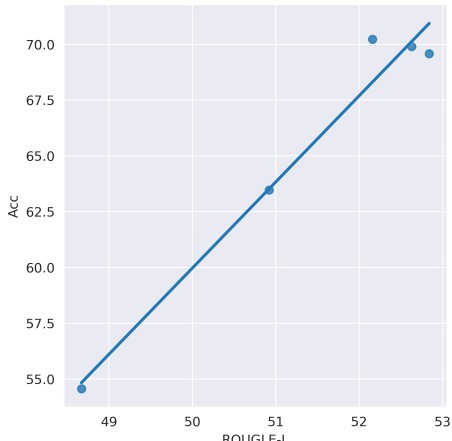

Figure 3: The linear regression between judgment accuracy and ROUGE-L. **Acc:** The correctness of judgment. **ROUGE-L:** The performance of downstream tasks.

training samples (84 different tasks with 60 training samples each). The entire template for generating new positive and negative examples has shown in the appendix (see Figure 6).

The performance with generated examples has been shown in Table 4. We find that TADIS still benefits the performance in both zero-shot and few-shot settings with generated examples, which improve 5.08 and 1.36 averaged ROUGE-L, respectively. However, the smaller model benefits more than larger model, which seems to be contrary to our previous experimental results. We suspect this

is because there are only 54% generated samples are valid as Self-instruct reports (Wang et al., 2023). Therefore, the larger model (3B) only improves the average ROUGE-L by 1.36 as it is misled by the partial wrong thinking labels. For the smaller model (770M), TADIS tends to induce the model learning the thinking format to elicit a better answer.

Table 4: The ROUGE-L results with generated examples by Self-instruct method (ChatGPT as the backbone; (Wang et al., 2023)) in zero-shot and few-shot settings on held-out dataset.

| Model Size | Testing Setting → Training Setting ↓ | Zero-Shot | Few-Shot | Avg RougeL |
|---|---|---|---|---|
| 770M | SuperNI(Few-Shot) | 23.08 | 40.54 | 31.81 |
| | TADIS | **32.62** | **41.16** | **36.89** |
| 3B | SuperNI(Few-Shot) | 36.38 | 43.09 | 39.73 |
| | TADIS | **37.95** | **44.23** | **41.09** |

Table 5: The ROUGE-L results for different stages in zero-shot and few-shot settings on held-out dataset. **Separated:** divide thinking and answering into two independent stages. **Combined:** thinking first, answering later.

| Model size | Stage | Zero-Shot | Few-Shot | Avg RougeL |
|---|---|---|---|---|
| 770M | Separated | **40.69** | 29.11 | 34.90 |
| | Combined | 33.58 | **46.66** | **40.12** |
| 3B | Separated | 42.82 | 43.40 | 43.11 |
| | Combined | **43.09** | **52.11** | **47.60** |

**Does the thinking and answering stages of TADIS can be separated?** Our ultimate goal is to make the model learn from examples and elicit a better answer. Therefore, we explored whether the same performance could be achieved by separating the thinking and answering stages of TADIS. Specifically, we separated a standard TADIS training sample into two sub-samples: evaluation instance (i.e., zero-shot) and the task of judging the correctness of provided examples. (i.e., judge whether these examples satisfy the requirements of task definition.) A concrete instance is shown in the appendix (see Figure 4). It's worth noting that we do not perform judgment tasks for each example but combine multiple examples together for judgment. This strategy can avoid causing an imbalance in the number of real tasks and judgment tasks, which could affect the performance of downstream tasks.

The results are shown in Table 5. We observe consistent performance drops when the thinking and answering stages are separated, which indicates the necessity of combining thinking and answering. The performance decreases by 5.22 and 4.49 average ROUGE-L with 770M and 3B model sizes, respectively. This is not surprising because it's the same for humans that we also think and then use the verification results to come up with better answers instead of separating them. Overall, combining thinking and answering stages can further elicit models for better answers.

**Does actions after thinking matter in TADIS ?** We performed ablation experiments to explore the importance of actions after thinking. Specifically, we directly remove actions in standard TADIS (i.e., "I should learn from correct examples and avoid mistakes in the wrong examples"). Table 6 depicts the performance with or without actions. We find that the performance without actions will drop by 1.61 and 0.82 ROUGE-L in zero-shot and few-shot settings, respectively, showing the indispensable of actions after thinking. In a nutshell, actions and thinking are equally pivotal in TADIS. Reminding the model of what actions it should take after thinking can elicit better answers. When both think-

Table 6: The performance without actions after thinking in TADIS. **w/o TA:** A standard SuperNI (few-shot) method without thinking and actions. **w/o TR:** We directly remove the text: *I should learn from correct examples and avoid wrong examples*

| Method | Zero-Shot | Few-Shot | Avg RougeL |
|---|---|---|---|
| TADIS | **43.09** | **52.11** | **47.60** |
| - w/o TR | 41.48 | 51.29 | 46.38 |
| | (-1.61) | (-0.82) | (-1.22) |
| - w/o TA | 38.50 | 51.08 | 44.81 |
| | (-4.59) | (-1.03) | (-2.79) |

ing and action are removed (w/o TA) and the correctness labels of the examples are specified in the prompt, the model in Wang et al. (2022) has lower performance on zero-shot and few-shot settings. Compared with TADIS, its ROUGE-L scores drop by 4.59 and 1.03 in zero-shot and few-shot settings, respectively.

Table 7: The performance (ROUGE-L) with different numbers of demonstration examples in zero-shot and few-shot settings. **N pos and M neg:** There are N positive examples and M negative examples in each training sample at most.

| Model | #Params | Testing Setting → Training Setting ↓ | Zero-Shot | Few-Shot | Avg RougeL |
|---|---|---|---|---|---|
| T5-Large-LM-Adapt | 770M | SuperNI(1 pos and 1 neg) | 33.30 | 45.08 | 39.19 |
| | | SuperNI(2 pos and 2 neg) | 30.75 | 45.82 | 38.28 |
| | | TADIS (1 pos and 1 neg) | **33.58** | **46.66** | **40.12** |
| | | TADIS (2 pos and 2 neg) | 28.66 | 45.85 | 37.26 |
| T5-XL-LM-Adapt | 3B | SuperNI(1 pos and 1 neg) | 38.54 | 51.08 | 44.81 |
| | | SuperNI(2 pos and 2 neg) | 35.72 | 49.64 | 42.68 |
| | | TADIS (1 pos and 1 neg) | **43.09** | **52.11** | **47.60** |
| | | TADIS (2 pos and 2 neg) | 38.92 | 51.41 | 45.17 |

**Do additional examples lead to further learning with TADIS ?**    Humans can improve their ability to complete downstream tasks by studying and learning from more demonstration examples. Therefore, we construct experiments to explore whether more examples lead to better performance. The results are shown in Table 7. The performance with different examples is extremely different from what we expected. More examples consistently lead to performance degradation with SuperNI and TADIS methods in zero-shot and few-shot settings. Actually, this conclusion is consistent with in-context learning that multiple demonstration examples are not necessarily better than one demonstration example (Chen et al., 2023a). Specifically, the 770M and 3B models dropped by 2.9 and 2.4 respectively. We suppose there are two reasons: (1) For the smaller model, more examples lead to more multiple types which increases the difficulty of learning format. (2) For the larger model, a pair of positive and negative examples is enough for model to learn the difference between them, and more examples will distract it. We will further explore the detailed reasons for this phenomenon in the future.

## 5    Conclusions and Future Work

In this paper, we propose TADIS, a method that can steer language models for Deep-Thinking about shown examples to elicit a better answer. The model is expected to learn from the example verification process, which encourages the model to have Deep-Thinking of examples instead of merely seeing. Finally, using the thinking as conditions will elicit models for a better answer. Experimental results on held-in and held-out datasets in both zero-shot and few-shot settings demonstrate the effectiveness of TADIS. In our preliminary experiment, TADIS is observed to improve the instruction tuning performance with examples created with Self-instruct method, which indicates a promising approach for instruction tuning in more diverse applications. Due to the limitation of computing resources, we do not experiment with larger models such as LLaMA(Touvron et al., 2023), we will scale our experiments in the future.

## 6    Limitations

The proposed TADIS method requires both positive and negative examples which are not readily available for many datasets. Although these examples can be created with Self-instruct method, the generated examples have to be carefully filtered and selected as only 54% generated samples are valid according to the observations in Wang et al. (2023).

## 7 REPRODUCIBILITY STATEMENT

The supplementary materials include the training and test data sets used in our experiments (generated through the SuperNI official code [2]) as well as the training code and scripts. We provide a detailed template for TADIS, Self-instruct and training with separate two Stages in Appendix A.1. Considering that reproducing results from scratch can be prohibitively expensive, we plan to publish our experimental results to promote reproducibility.

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

# A  APPENDIX

## A.1  DIFFERENT DATA FORMAT

**Data Template for TADIS.**    Our proposed TADIS method takes the task definition, examples and instance input as the prompt. The model first generates the verification result and corresponding action of the provided examples, then outputs the reply for the instance input.

---

**Task Definition:** {{definition}}
**Example 1 -**
-        **Input: {{exp.input}}**
-        **Output: {{exp.output}}**
**Example 2 -**
-        **Input: {{exp.input}}**
-        **Output: {{exp.output}}**
**Evaluation Instance -**
-        **Input: {{exp.input}}**

- - - - - - - - - - - - - - - - - - - - - - - - - - - - - - - - - - - - - - - -

**Thinking -**
-        **thinking and actions: {{Example 1 is correct/wrong and example 2 is correct/wrong. I should learn from correct examples and avoid the mistakes in these wrong examples.}}**

- - - - - - - - - - - - - - - - - - - - - - - - - - - - - - - - - - - - - - - -

**Answering -**
-        **Output: {{exp.output}}**

---

Figure 4: The data template used for TADIS method.

**Data Template Used for Training with Separate Two Stages.**    When the model is trained with separate thinking and answering stages (see **Does the thinking and answering stages of TADIS can be separated?** in Section 4.3), the model will only verify the correctness of provided examples in the thinking subtask.

---

**Task Definition:** {{definition}}
**Example 1 -**
-        **Input: {{exp.input}}**
-        **Output: {{exp.output}}**
**Example 2 -**
-        **Input: {{exp.input}}**
-        **Output: {{exp.output}}**
**Judge whether each example conforms to the task definition.**
**Prediction: {{Example 1 is correct/wrong and example 2 is correct/wrong.}}**

---

Figure 5: The data template used for training with two separated stages.

**Data Template for Generating Examples with Self-Instruct.** When generating positive and negative examples with Self-instruct method, we randomly select four pairs of positive and negative examples from the example pool as in-context learning examples. The ChatGPT will generate a positive and negative example pair based on the prompt. We use the data template as follows:

---

**Few-Shot:**
**Task Definition:** {{definition}}
**Positive Example**
-        **Input: {{exp.input}}**
-        **Output: {{exp.output}}**
**Negitave Example**
-        **Input: {{exp.input}}**
-        **Output: {{exp.output}}**
**......**

- - - - - - - - - - - - - - - - - - - - - - - - - - - - - - - - - - - - - - - - - -

**Generated Examples:**
**Positive Example**
-        **Input: {{gen.input}}**
-        **Output: {{gen.output}}**
**Negitave Example**
-        **Input: {{gen.input}}**
-        **Output: {{gen.output}}**

---

Figure 6: The data template for creating positive and negative examples with Self-instruct method.

## A.2  TRAINING DETAILS

For both the `T5-Large-LM-Adapt` (770M) and `T5-XL-LM-Adapt` (3B) models, we finetuned them for five epochs, with batch size as 64. We use Adam optimizer with $\beta_1 = 0.9$, $\beta_2 = 0.999$. The linear learning rate scheduler starts from $2 \times 10^{-4}$, then decays to 0. We tried to use the commonly used learning rate $2 \times 10^{-5}$, but found that the model did not converge well. Therefore, we use a large learning rate $L_r = 2 \times 10^{-4}$ which is also same as that in Kung & Peng (2023). All of experiments are run on $8 \times 4090$ GPUs with 24G memory. The models are trained with `Huggingface Transformers` toolkit. The maximum input length is 1024 and the maximum output length is 128. This reproduces the results in Wang et al. (2022).

For the inference of TADIS in the zero-shot setting, although no examples are provided, we randomly add the thinking results about the imagined examples to align with the format used during training. We apply position shuffling and multiple example sampling strategies to prevent the model from directly memorizing examples instead of thinking about them. Specifically, we construct four different types of input data: (1) **Without examples:** A training sample without any example. (2) **Only negative examples:** A training sample with only negative examples.(3) **Only positive examples:** A training sample with only positive examples.(4) **Mixing examples:** A training sample with both positive and negative examples.

During training, we sample examples to enable the overall input data to meet with the maximum input length limit. Specifically, for each input data sample, we first add *task definition* and *evaluation instance*. Then add *example 1* and *example 2* respectively. If the number of tokens is greater than max length at any stage, we will stop. In this way, we could obtain multiple types of provided examples. The proportions of 'without examples', 'only positive examples', 'only negative examples' and 'mixing examples' are 2.9%, 6.3%, 0.5% and 90.2% respectively in our constructed training data.

