# OpenReview forum: "TADIS: Steering Models for Deep-Thinking about Demonstration Examples"
_ICLR.cc/2024/Conference — ICLR 2024 Conference Withdrawn Submission_

### Official Review · Reviewer_B4dK · 2023-10-28

**Soundness:** 2 fair
**Presentation:** 3 good
**Contribution:** 2 fair
**Rating:** 3
**Confidence:** 2

**Summary:**

This paper addresses the task of instruction tuning where a large language model (LLM) can leverage additional context (task definition, examples) to achieve strong zero- and few-shot performance on several downstream tasks. The paper highlights prior work which suggested that models are learning formatting from the examples, rather than the actual mapping. This is supported by the finding that models perform comparable even if provided with incorrect, but properly formatted, examples. The paper proposes improving a previous approach by training the model model to reflect on the accuracy of the examples before answering. The proposed approach is shown to outperform the previous approach on a common benchmark. Further experiments are conducted to analyze the model's performance.

**Strengths:**

- The model shows performance over the standard approach (SuperNI) with
- The figures nicely illustrated the setup and model performance.
- The paper includes a lot of analysis which is framed as separate concrete questions, allowing the reader to easily comprehend the point being made.
- The paper includes several analysis experiments that analyze different facets of the model.

**Weaknesses:**

- The paper is motivated by the findings of Kung and Peng (2023) that analyzed instruction-tuning models that learn from examples. They found that models seem to learn the format of the examples, rather than the mapping in those examples. This results in models achieving similar performance whether they are provided with accurate or inaccurate examples. While the paper aims to tackle this problem, it seems that TADIS exhibits the same issue. In particular, Table 3 shows that the models trained with true or flipped thinking results achieve comparable performance with only slight performance drops. Furthermore, across all models, the performance is comparable between receiving correct, flipped, or random thinking results. The only major drop happens for the larger model trained with random thinking results. The paper explains this result by saying that TADIS helps more during training than testing, however, if the model is relying on the "thinking" result, it should be affected at inference time as well. As a result, it is unclear to me that the problem identified in the paper has been addressed.

- Following on the previous point, Kung and Peng (2023) showed that models perform comparably when provided with accurate or delusive instructions, however, the comparsions done in Table 3 is when the thinking results is flipped. It is unclear if the model is ever evaluated on delusive examples.

- The paper shows a correlation between judged accuracy of statements and the generated score, and uses that to support the claim that forcing models to "judge the correctness of [the] provided examples could steer them towards deep thinking." However, it is completely expected that those two values would be correlated even if there was no causal link between learning to judge the correctness of the statements and learning to generate more accurate responses. To support the claim that judging accuracy improves model performance, one needs to evaluate a setting where you evaluate the correlation between those two numbers without having the model learn to judge the accuracy. For example, getting the model to judge the accuracy of the statements without training or doing it after the model generates the answers.

- The paper uses the findings of Kruger and Dunning (1999) to motivate their problem by connecting it to the findings of Kung & Peng (2023). Kruger and Dunning (1999) studied the relationship between skill and self-assessment of performance, and found they people over-estimated their abilities. Kruger and Dunning argued that poor performance and miscalibrated assessment were both caused by low competency; ie, one's ability to perform a task affects their ability to assess how well that task is done. This is very different from the findings of Kung & Peng (2023) where the model simply learns a different thing from what is learned. Furthermore, there is no self-assessment or estimation of the accuracy of one's predictions for both the proposed model or in Kung & Peng (2023). While the "thinking" stage does include an assessment of the accuracy of the inputs, this assessment is of the inputs not the model's predictions or overall model capability. As a result, I do not see how Kruger and Dunning is relevant to this work.

- The paper includes a lot of anthropomorphizing of different components of the model (eg, thinking, seeing, acting) which sometimes does not indicate different model behaviors or capabilities. I provide two concrete examples of this below. I think this is problematic because it (a) ascribes behaviors, capabilities, and agency to the model which do not seem to be there; (b) drowns the technical explanations with metaphors which confuses the point for no clear reason. One possibility is that those terms are all aliases/synonyms for something more technical. In that case, I think the paper woould be much stronger and clearer if the exact capability is described.
    - The method name is an example of this as it implies the model is thinking instead of seeing, however, it is unclear why standard in-context learning is only seeing, while an extra prompt should be categorized as thinking.
    - While I understand why it might be good to differentiate the "thinking" and "answering" losses as they deal with two different things (generating accuracy labels for the inputs vs. generating the answer), the method section is full of words that indicate the same thing, those include "deep thinking", "thinking", "answering", and "generating actions."

- Typo in the second paragraph on page 8: The paper references Figure 4 as an example of separating the two stages, when it should be figure 5.

**Questions:**

- Do any of the experiments evaluate the model's performance on delusive examples? If so, could you please indicate which ones?
- It seems to me that the results in Table 3 suggest that the model is insensitive to the thinking results, which contradicts the explanation provided for how the model works. Do you think I misunderstood this? if so, could you explain how the thinking result is important to the prediction task if flipping the thinking results doesn't affect performance?
- Could you explain the distinctions between the terms used to describe model behavior (seeing, thinking, acting, etc)?
- Could you please elaborate on the connection to Kruger and Dunning (1999)?
- I was curious why ROUGE-L was used as a metric. It was suggested that this metric corrrelates well with accuracy for classification tasks and evaluation, but can this metric discriminate between accurate and inaccurate statements? If possible, what is the ROUGE-L score between the correct and delusive examples? or between the ground-truth and flipped thinking results?

---

### Official Review · Reviewer_Fers · 2023-10-30

**Soundness:** 2 fair
**Presentation:** 2 fair
**Contribution:** 2 fair
**Rating:** 3
**Confidence:** 4

**Summary:**

The paper presents an instruction tuning technique for large language models (LLMs). The goal is to fine-tune a pre-trained LLM to improve its generalization on unseen tasks given task definition and some working examples. The proposed technique consists of two stages; In the "thinking" stage, the model learns to classify the given examples as positive or negative. In the "answering" stage, the prediction of task output is conditioned on those labels and an action to facilitate the reasoning. On the SuperNI-v2 benchmark, the proposed method outperforms SuperNI in both zero-shot and few-shot settings.

**Strengths:**

- This is a timely study on a trendy research topic (LLM instruction tuning). The paper aims to address the concern that LLMs learn to emulate the output format rather than follow the instructions in instruction tuning. The key observation is that instruction tuning can achieve competitive results even when the examples have incorrect input-output mappings. To this end, the paper introduces a new instruction tuning technique which routes the reasoning of an LLM through an intermediate step, that is, checking the correctness of the provided examples, with the intuition that it can facilitate instruction understanding.

- The approach is simple and effective. It breaks instruction tuning into two stages, with simple learning objective and inference procedure for each stage. Despite its simplicity, the method performs well on a recent task generalization benchmark (SuperNI-v2). It outperforms the baseline in both zero-shot and few-shot settings, and shows improved performance with larger LLM backbones.

**Weaknesses:**

- The main claim of the paper is that the proposed method promotes "thinking" on the examples, which informs subsequent "answering". More precisely, "thinking" amounts to classifying the examples as positive or negative, and the way "thinking" influences "answering" is through conditioning the task output on the classification labels. However, the results in Table 3 challenge this claim. At inference time, whether or not an example is correctly labelled has absolutely no impact on model performance. That is, despite being fed the "thinking" result, the model chooses to ignore it when predicting the task output. This leads me to think that the proposed method does not work as the authors claim, and it does not solve the problem that initiated the work.

- I encourage the authors to further study the source of performance gain. According to Table 3 and Figure 3, my hypothesis is that example classification is an effective auxiliary task even though it does not encourage an LLM to follow instructions. Further, I suspect that the specific instruction format (especially with the action text) boosts performance (Table 6 provides some evidence).

- Comparing results in Table 2 (first row) and Table 4, it is clear that LLMs fine-tuned using the proposed method on synthetic examples perform worse than zero-shot SuperNI (i.e., without using examples). This challenges the second contribution of the paper, which says the method can scale up instruction tuning by synthesizing examples for a large number of tasks. In the case where examples are lacking, one would rather choose not using examples (i.e., zero shot) than synthesizing examples for few-shot instruction tuning.

**Questions:**

- While I am not fully convinced that the method works as expected, I am impressed by the performance gain given the simplicity of the method. I encourage the authors to design additional ablation experiments to better understand the source of performance gain, and modify their claims accordingly.

- Reading between the lines, I have the impression that the "answering" stage always sees the same piece of action text. Then why does the "thinking" stage need to predict the action text?

---

### Official Review · Reviewer_B5Pw · 2023-10-31

**Soundness:** 2 fair
**Presentation:** 1 poor
**Contribution:** 2 fair
**Rating:** 3
**Confidence:** 2

**Summary:**

The paper proposes a prompt augmentation and training scheme to improve LLM in-context learning (from few examples). The core idea is to follow the examples with an immediate prompt of a form like "examples 1,2,3 are correct and examples 4,5,6 are wrong -- I should learn from correct examples and avoid incorrect ones".
Experiments are trained/evaluates on SuperNI-V2 using two sizes of T5 (T5-Large-LM-Adapt and T5-XL-LM-Adapt).

**Strengths:**

The paper chooses a good dataset, the method is somewhat clear, and there are some reasonable comparisons to baselines (SuperNI).

The problem setting is an important one, and the flavor of the solution is reasonable -- similar to things like "let's think step-by-step".

### Experiments
I like the experiment training and testing with different settings of random/flipping of the labels. The results are reasonable that the random shuffling of correctness harms the models performance. The model learns to mostly compensate for flipped labels (though the authors ignore this, and focus on how the performance is just a little bit worse than the original setting.

**Weaknesses:**

Overall the paper is a bit hard to follow. At a high level, this is a relatively incremental change to an existing approach, evaluated on a single dataset. The results are mixed: generally a small improvement -- though not always.

### Presentation
The paper is a bit hard to follow. I try my best to ignore syntax -- many of the best researchers in the field are nonnative english speakers. The presentation is hard to follow for other reasons -- e.g. section 2.1 is a single paragraph that takes up more than half a page.

Some parts of the method are not defined, e.g. Actions $A_e$  in section 3. What exactly are the actions?

### Experiments
1. Evaluation is only on two relatively small variants of T5. The results seem mixed for the smaller model, though a bit better for the 3B model. It seems like the way to evaluate this would be to train larger models
2. The lack of error bars makes it hard to judge the significance of the performance deltas
3. I found the ablation study a bit hard to understand -- so if (w/o TR) is just removing that one line, why is the drop-off so much larger than removing the whole thinking section? Shouldn't the model learn to compensate for the lack of prompt? Or is this evaluation removing the prompt at inference for a model trained with the full prompt (this taking the model OOD)

**Questions:**

1. Is the Thinking loss just a loss that requires the model to begin with "I should learn from correct examples and avoid mistakes in the wrong examples”?
2. It is indeed a bit counterintuitive that the models do worse with more examples (2 positive and 2 neg, vs 1 pos and 1 neg). I like that the authors included this experiment -- I would ask for it if it weren't there. Still, I feel like this result needs some explaining, since the idea of this method is to make the model better able to leverage in-context learning, and the drop-off with more examples actually seems greater with TADIS vs. w/o it.

---

### Official Review · Reviewer_8SUE · 2023-10-31

**Soundness:** 3 good
**Presentation:** 2 fair
**Contribution:** 3 good
**Rating:** 6
**Confidence:** 3

**Summary:**

This paper introduces TADIS, a novel method for fine-tuning LLMs in few-shot settings and improving their performance in natural language tasks. TADIS proposes to modify the typical approach of providing positive and negative prompts in few-shot examples with an additional classification (referred as 'deep-thinking' in this paper) step to reason the truthfulness of the examples, before predicting a final answer for the concerned task. TADIS was tested in zero-shot and and few-shot settings and was shown to outperform baseline few-shot fine-tuning in the ROUGE-L metric, for both in-domain and out-of-domain tasks.

**Strengths:**

- Outperforms baseline work with a simple method that is applicable to many tasks.
- Extensive exploration and experiments for multiple variants of the method, some based on real use-cases such as noisy generated task examples.
- Comprehensive Ablation Study and Discussion

**Weaknesses:**

- **Limited novelty over recent prompting work**: the proposed methods is conceptually similar to Chain-of-Thought prompting, falling under the same theme of adding intermediate reasoning steps before making a final prediction. Nevertheless, the proposed method might be more widely available to existing data given that it does not have additional requirement beyond truthfulness labels for the examples

- **Requires re-training of the concerned models**: which could be costly given the targeted models are LLMs, when compared to the few-shot prompting paradigm.

- **Marginal improvement over baselines**: Only few-point improvement in ROUGE-L score is observed over baselines, in which a larger model may demonstrate greater improvements for the proposed method.

- **Some overly broad claims in the presentation of the paper**: statements such as steering models to perform 'deep-thinking' might require more extensive evidence and prior literature to support that the LLMs are actually 'thinking'.

**Questions:**

As the author(s) claim that TADIS elicits deep-thinking behaviour in models, I wonder if the author(s) have explanations on why the "Random" test setting can improve over Zero-shot test setting for all methods in Table 3? I would expect that the model should not have gained any new knowledge from Random labels, and would perform similarly to zero-shot. While the author(s) have explained this behavior for the training setting, I would like to know why does this not also apply to testing?

Based on the above question, I would also expect the that ground-truth testing setting to perform better than the random testing setting in Table 3, and would like to the author(s) to discuss these results further. It might also be helpful to show some qualitative results comparing between the generated outputs from various methods.